# Phylogenetic Distribution of WhiB- and Lsr2-Type Regulators in Actinobacteriophage Genomes

Vikas Sharma,[a] Aël Hardy,[a] Tom Luthe,[a] Julia Frunzke[a]

[a]Institute of Bio- and Geosciences (IBG-1) Biotechnology, Forschungszentrum Jülich, Jülich, Germany

**ABSTRACT** Viruses that infect different actinobacterial host species are known as actinobacteriophages. They are composed of highly divergent and mosaic genomes due to frequent gene exchange between their bacterial hosts and related viral species. This is also reflected by the adaptive incorporation of host transcription factors (TFs) into phage regulatory networks. Previous studies discovered Lsr2-type and WhiB-type regulators encoded by actinobacteriophage genomes. However, limited information is available about their distribution, evolution, and impact on host species. In this study, we computationally screened the distribution of known bacterial and phage TFs inside 2951 complete actinobacteriophage genomes and identified 13 different TF domains. Among those, WhiB, Lsr2, MerR, and Cro/CI-like proteins were widespread and found in more than 10% of the analyzed actinobacteriophage genomes. Neighboring genomic context analysis of the *whiB* and *lsr2* loci showed group-specific conservation of gene synteny and potential involvement of these genes in diverse regulatory functions. Both genes were significantly enriched in temperate phages, and the Lsr2-encoding genomes featured an overall lower GC content. Phylogenetic analysis of WhiB and Lsr2 proteins showed the grouping of phage sequences within bacterial clades, suggesting gene acquisition by phages from their bacterial host species or by multiple, independent acquisition events. Overall, our study reports the global distribution of actinobacteriophage regulatory proteins and sheds light on their origin and evolution.

**IMPORTANCE** Actinobacteriophages are viruses that infect bacterial species of the diverse phylum of Actinobacteria. Phages engage in a close relationship with their bacterial host. This is also reflected by the adoption of genetic material from their host and its incorporation into phage regulatory circuits. In this study, we systematically searched the genomes of actinobacteriophages for the presence of transcription factor domains. We show that proteins belonging to the regulator families of WhiB and Lsr2 belong to the most abundant regulatory proteins encoded by actinobacteriophages. Further phylogenetic analysis shed light on their origin and evolution. Altogether, this study provides an important basis for further experimental investigation of their role in the coordination of the phage life cycle and their interaction with the host regulatory network in this important bacterial phylum.

**KEYWORDS** actinobacteriophages, transcriptional regulators, comparative analysis, phylogeny, WhiB, Lsr2, Actinobacteria, bacteriophages, phylogenetic analysis

Address correspondence to Vikas Sharma, v.sharma@fz-juelich.de, or Julia Frunzke, j.frunzke@fz-juelich.de.

Actinobacteriophages are viruses that infect hosts of the bacterial phylum Actinobacteria (1). This phylum includes a variety of Gram-positive bacteria that are of high medical, ecological, and biotechnological relevance. It covers important human pathogens from the genera of *Mycobacteria, Corynebacteria*, and *Streptomyces* as prominent inhabitants of the soil and antibiotic producers in addition to the nonpathogenic bacterium *Corynebacterium glutamicum*, which is one of the most important organisms in the biotechnology platform (2).

The continuous effort of research- and education-oriented programs such as the 'Phage Hunters Integrating Research and Education' (PHIRE) and 'Science Education Alliance-Phage Hunters Advancing Genomics and Evolutionary Sciences' (SEA-PHAGES) has contributed to the discovery and sequencing of a variety of actinobacteriophage-infecting species of different actinobacterial genera (3, 4). As of the spring 2021, the actinobacteriophage database (https://phagesdb.org) comprised of more than 18,000 isolated phages and more than 3500 complete genome sequences. While >2000 mycobacteriophages that are listed in the database were predominantly isolated using the host *Mycobacterium smegmatis* mc²155, the phages of *Gordonia, Arthrobacter, Microbacterium, Actinoplanes, Propionibacterium, Rhodococcus, Tsukamurella, Tetrasphaera, Corynebacterium, Brevibacterium, Rothia*, and *Streptomyces* were isolated on different strains and species (1, 3, 5).

This wealth of information unraveled the substantial genomic diversity of phages that infect Actinobacteria. Genomes of these phages include a significant amount of viral 'dark matter', with more than 70% of their genes encoding proteins of unknown function (1, 6, 7). Moreover, all isolated actinobacteriophages possess a linear, double-stranded DNA genome that ranges in size from 14 to 194 kbp and belong to the viral families of *Siphoviridae, Myoviridae*, and *Podoviridae* (1, 4, 8). Only a few examples of nontailed *Tectiviridae* phages that feature lipid-containing virions have been described (9). Actinobacteriophage genomes comprise a highly mosaic genome architecture, which is probably the result of nonhomologous recombination events rather than sequence-directed processes. These processes are likely facilitated by systems that catalyze nonhomologous end joining (NHEJ) or may also involve Ku-like proteins, which have been found in several phages (10). Investigation of the gene exchange rate between these phages revealed a high gene content flux within temperate phages compared to virulent phages (8).

The relationship between these phages was established based on overall pairwise nucleotide sequence similarity and shared gene content, which divides actinobacteriophage genomes into more than two dozen distinct clusters and over 100 subclusters. A cluster comprises a group of phages that shares at least 35% of genes with at least one other member of the cluster. Phages without close relatives are assigned as singletons (11, 12). A similar phylogenetic pattern was observed by using genome-wide, shared protein family (Pfam) domains content-based clustering. The resulting analysis also delineated clades in close agreement with already defined clusters and proposed an alternative approach to classifying actinobacteriophage genomes (5). However, these data are markedly affected by sampling bias (e.g., with more than 1800 phages infecting *M. smegmatis*) and the true picture likely represents a continuum of diversity.

Actinobacteriophages can be grouped into one of two different lifestyles. Obligatory lytic phages use their host as a 'virion production factory', which typically leads to cell lysis and death of the host cell. In contrast, temperate phages follow one of two life cycle paths following infection of the host cell. They will either follow the lytic path to replicate and produce new viral particles, or they will follow the lysogenic path and establish a long-term association with their host cell as a prophage by efficiently silencing the expression of lytic genes (13). Stressful environmental conditions may trigger the induction of the prophage, which results in the entry into the lytic cycle. An intricate genetic switch encoded by the phage system accomplishes this decision. *Escherichia coli* phage lambda represents the canonical example of developmental decision making (14). Here, the lysis-lysogeny decision is defined by the CI and Cro regulators in a bistable genetic switch. While regulators of the Cro/CI family are ubiquitous in actinobacteriophages, the overall regulatory equipment largely differs from that of the extensively studied phages that infect Proteobacteria.

Recent studies described the importance of the role of the Lsr2-like protein CgpS for maintaining the lysogenic state of the large prophage CGP3 of *C. glutamicum* (15). Lsr2-like proteins are highly conserved among members of the phylum of Actinobacteria in which they are involved in the control of horizontally-acquired ('xenogeneic') genomic regions, including virulence gene clusters, cryptic prophages, and cryptic metabolism (15–18). The

Lsr2-like protein CgpS was the first phage-encoded xenogeneic silencer protein to be described and was found to bind AT-rich regions within the CGP3 island in which it tightly silences phage gene expression (19). Lsr2-like proteins were later found in additional phage genomes, but a comprehensive analysis is currently missing (20).

A further class of regulatory proteins that are exclusively found in Actinobacteria are the WhiB-like proteins. Sequencing of a multitude of actinobacterial genomes revealed the prevalence of these regulators within this phylum, where they carry out important functions like virulence, morphological differentiation, stress response, and antibiotic resistance (21, 22). WhiB-like proteins were also described in phages infecting *Mycobacteria* species. WhiB of phage TM4 was shown to inhibit the expression of the host *whiB2* gene resulting in morphological changes upon expression (23). WhiB of phage Tweety was suggested to be targeted by prophage-mediated defense mechanisms (24). Yet, the functions of WhiB-like proteins in the phage life cycle remain largely enigmatic.

The adoption of prevalent actinobacterial regulators by actinobacteriophages illustrates the intricate link between the phages and their host and raises questions about their evolution and impact on the phage life cycle. In this study, we systematically analyzed the genomes of actinobacteriophages with respect to their regulatory equipment. This approach revealed proteins of the WhiB family as the most abundant regulators of actinobacteriophages, followed by members of the Cro/CI, MerR, and the Lsr2-like family. Comprehensive analyses that focus on WhiB- and Lsr2-like genes showed cluster-specific distributions and conserved gene synteny. Further phylogenetic analyses suggested multiple independent gene acquisition events by actinobacteriophages from the respective host species.

## RESULTS AND DISCUSSION

**Distribution of bacterial transcription factor domains in actinobacteriophages.** To explore the distribution of bacterial and phage transcription factors (TFs) in actinobacteriophages, we initially performed a genome-wide search of conserved protein domains obtained from 2951 complete phage genomes (Table S1) in the NCBI Conserved Domain Database (CDD) using RPS-BLAST (25). A search with known bacterial and phage transcription factor domains obtained from the P2TF (26), Pfam (27), and Phamerator databases (28) revealed 36 different major TF domains that were present in at least 10 or more of the 1702 actinobacteriophage genomes (Table S2). Twenty-eight out of the thirty-six identified domains are completely or partially overlapping within the corresponding genome coordinates but belong to different source databases (Pfam, Clusters of Orthologous Genes [COG], Simple Modular Architecture Research Tool [Smart], and The Institute for Genomic Research [TIGR]). Consequently, these identical domains were collapsed, and the list of actinobacteriophage-encoded TF domains was reduced to 13 unique TF domains whose distributions ranged from 0.4% to 23% within the analyzed genomes (Fig. 1 and Table S2). Interestingly, the WhiB, Cro/CI-type, MerR, and Lsr2 domains were the most abundant and present in more than 10% of the analyzed phage genomes. However, instead of cluster bias, these identified domains were prevalent in different groups of phages. Moreover, their distribution appears to not be lifestyle specific because these domains are encoded by temperate and virulent phages.

Next, we investigated the co-occurrence of different TF domains to understand if they are functionally linked or involved in the same regulatory circuit. The most striking result of this analysis was a negative correlation between the Cro/CI and Lsr2 domains, suggesting that phage genomes rarely share these two types of repressors (Fig. S1).

**Distribution of *whiB* and *lsr2* genes in actinobacteriophage clusters.** We focused our analysis on WhiB and Lsr2 regulators, which showed a negative correlation with the well-known Cro/CI-like proteins and were among the most abundant regulators found in actinobacteriophages (Fig. 1). To understand their evolutionary histories, an unrooted phylogenetic tree was constructed based on genome-wide k-mer frequency profiles clustering using 2951 complete actinobacteriophage genomes. The resulting phylogenetic tree delineated the phages into multiple distinct clusters or groups (Fig. 2) that showed high

Microbiology Spectrum

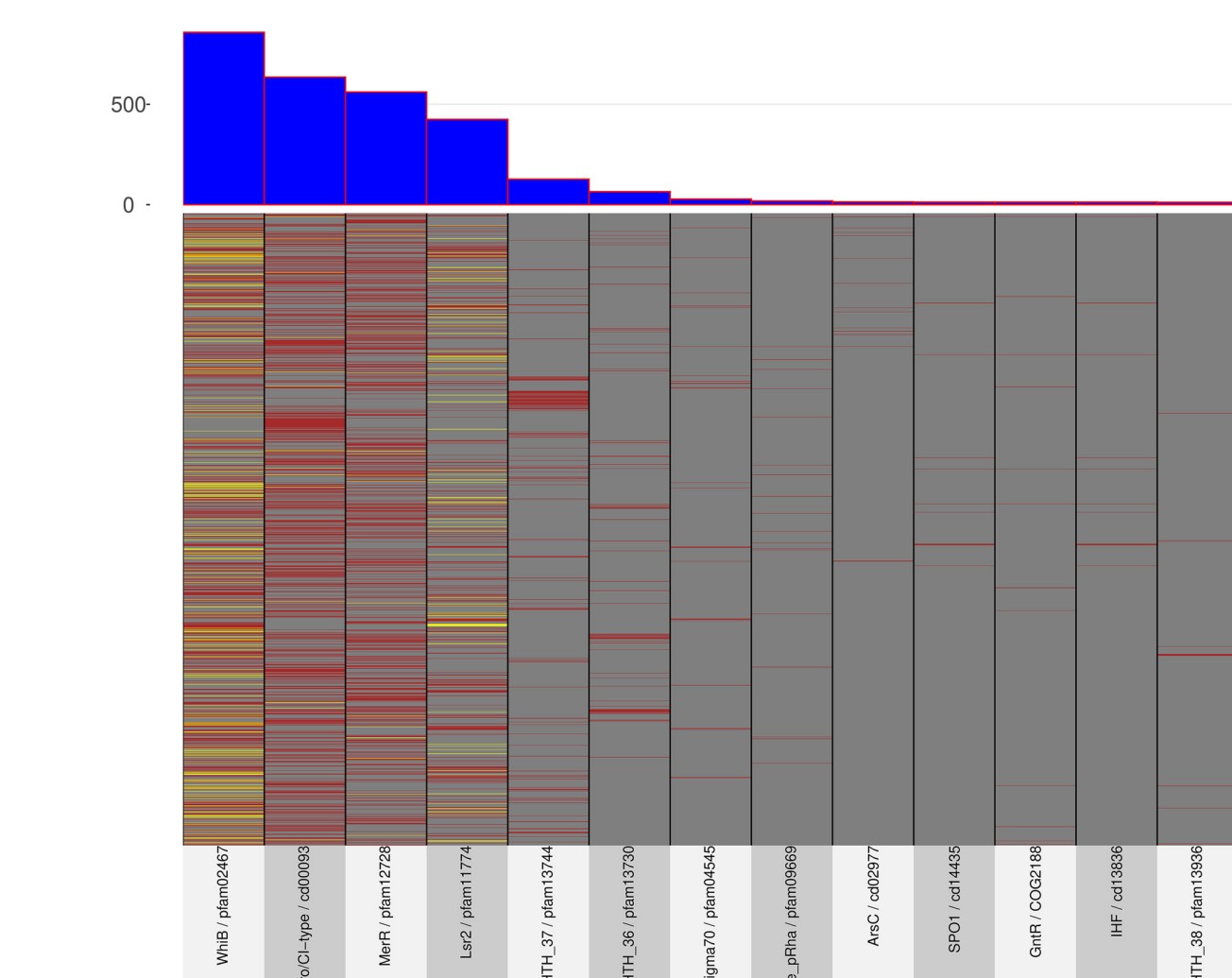

**FIG 1** Distribution of 13 transcription factor domains across 1702 actinobacteriophage genomes. The heatmap indicates the domain count (1 = red; 2 = yellow) per phage genome. The bar plot above the heatmap presents the overall domain abundance based on the total count of Pfam domains across the genomes.

congruence with the allocated clusters obtained from the PhagesDB database (https://phagesdb.org) (4). The phylogenetic mapping of the WhiB and Lsr2 protein domains on the actinobacteriophage tree revealed a nonrandom distribution across the known phage clusters. Except for a few clusters, individual proteins mostly showed a clade-specific distribution pattern, indicating that there was no biological link between the two proteins, which was in line with our correlation analysis (Fig. S1). Additionally, the phage genomes that encode more than one copy of the protein domains also fall within unique and specific clusters, suggesting multiple, independent gene transfer events following a common origin and evolution from group-specific ancestral viruses. Besides that, the level of conservation between the WhiB- and Lsr2-encoding phages were calculated using a pairwise genome comparison. The resulting analyses, according to the known clusters, reflected the conservation level. As expected, phages belonging to the singleton cluster showed a high variation in comparison to other clusters, suggesting the involvement of genes in diverse functions (Fig. S2). In contrast, a high conservation – as observed for most clusters – suggests that these regulators are playing critical roles in the life cycles of that particular group of phages.

**WhiB-type proteins represent the most abundant regulators in actinobacteriophages.** Remarkably, our analysis revealed that WhiB-type proteins were the most abundant regulator class in actinobacteriophages followed by the Cro/CI-type

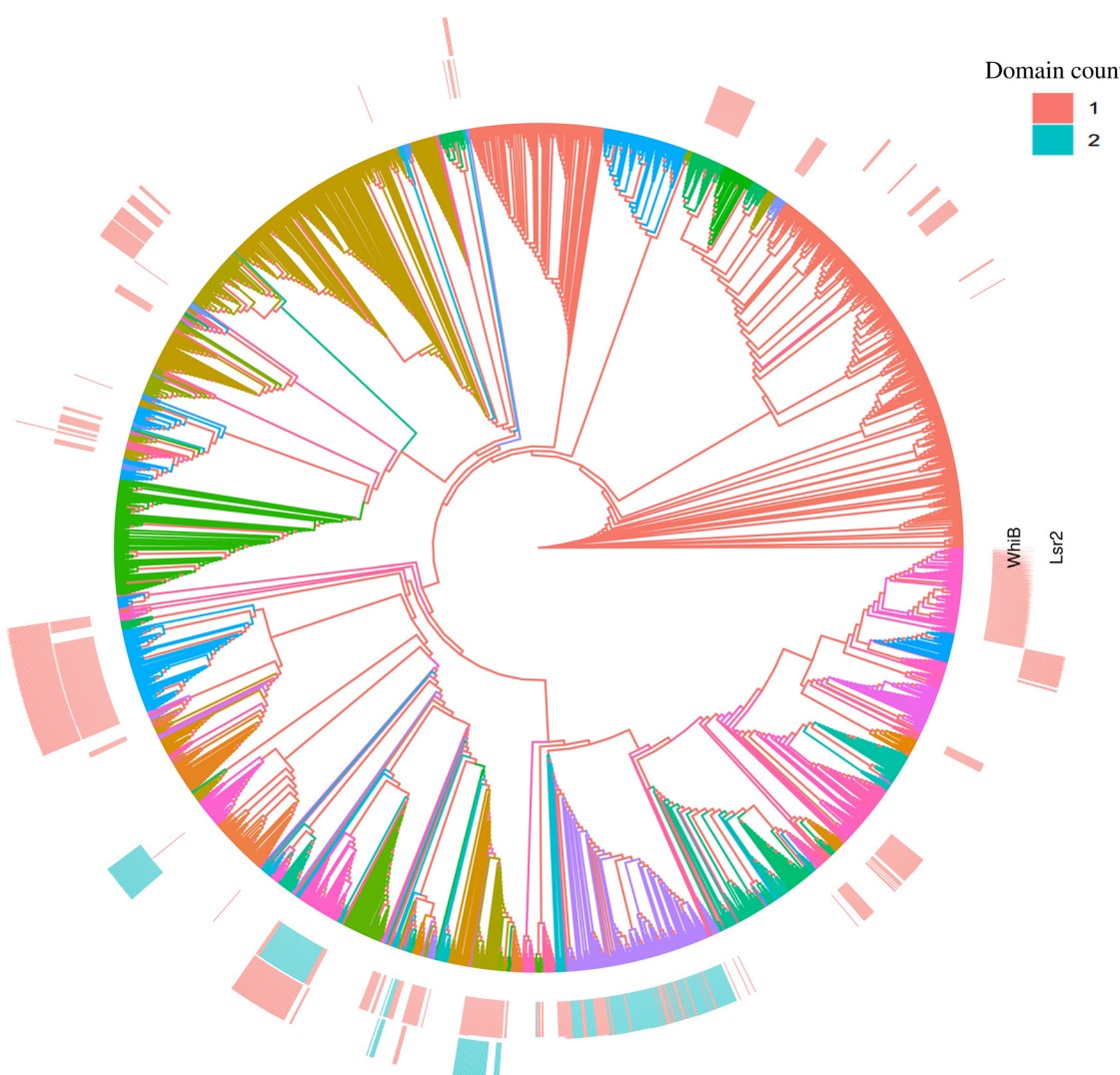

**FIG 2** Lsr2 and WhiB are nonrandomly distributed among actinobacteriophage genomes. The k-mer clustering-based phylogenetic tree is based on complete nucleotide sequences from 2951 actinobacteriophage genomes. Sequences are color-coded according to already assigned clusters information obtained from PhagesDB (4). The concentric heatmaps represent the Pfam domain counts (1 = green; 2 = red) of WhiB (inner ring) and Lsr2 (outer ring) of the corresponding genomes.

regulators and MerR- and Lsr2-like proteins. To provide insight into their potential role in the phage life cycle, further investigations were focused on phage-encoded WhiB- and Lsr2-like proteins. A systematic tBLASTn similarity search was conducted to detect all sequence homologs among the 2951 actinobacteriophage genomes using the query protein sequence library of the *whiB* and *lsr2* genes separately and a custom-designed pipeline. The resulting similarity searches revealed the presence of *whiB* and *lsr2* genes in 24% (711/2951) and 11% (354/2951) of the total analyzed actinobacteriophage genomes, respectively. Compared to the protein domain-based analysis, the BLAST-based similarity searches retained a few more sequence homologs of both genes (Table S2), which was probably due to parameter and methodological differences between the two approaches.

Overall, we identified more than >800 putative *whiB* genomic loci within 8 out of 13 different host genera (*Arthrobacter*, *Corynebacterium*, *Gordonia*, *Microbacterium*, *Mycobacterium*, *Propionibacterium*, *Rhodococcus*, and *Streptomyces*, Fig. 3A). *Tsukamurella* phages (1 *whiB* gene found in 2 phage genomes) were not included in the analysis due to the small dataset. *Mycobacterium* stands among the largest group of WhiB-encoding phages (with 492

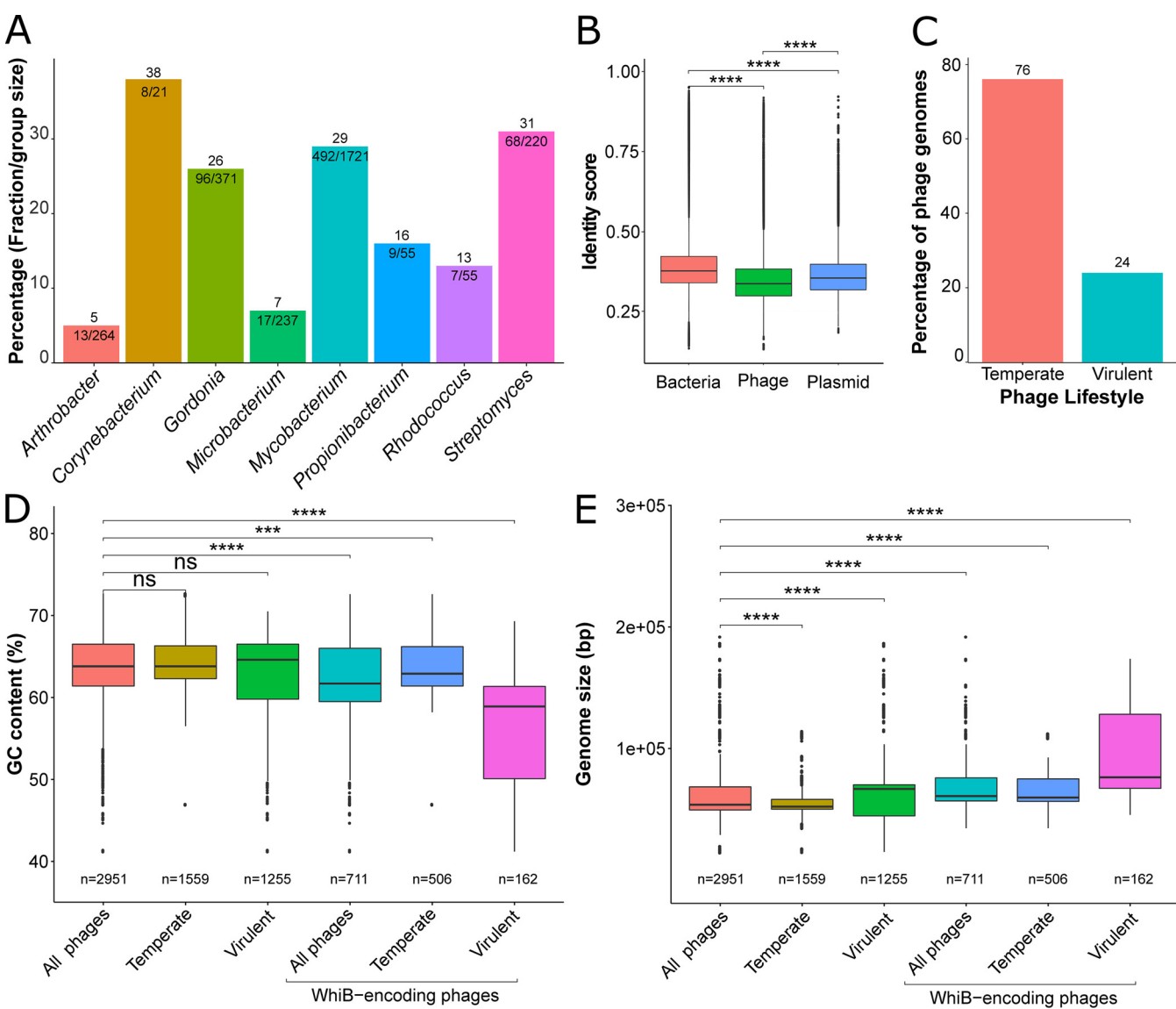

**FIG 3** WhiB-type proteins represent the most abundant regulators in actinobacteriophages. (A) Bar plot representing the actual count and proportion of WhiB-encoding phages according to host genus. (B) Global pairwise identity between WhiB amino acid sequences encoded by phage, bacterial, and plasmid phage genomes. (C) Distribution of WhiB-encoding phages is displayed according to phage lifestyle (temperate and virulent actinobacteriophages). (D) Box plot presenting the GC content of actinobacteriophages depending on their lifestyle and the presence of a *whiB* locus. (E) Box plot presenting the genome size of actinobacteriophages depending on their lifestyle and the presence of a *whiB* locus. Level of significance is indicated based on the *P value* (ns, not significant; ***, *P* < 0.001; ****, *P* < 0.0001).

representatives), but this is also a result of the high number of *Mycobacterium* phage genomes (>1700) in the actinobacteriophage database (4). The normalized percentage of WhiB-encoding genomes obtained by considering the overall group size of the corresponding host genus showed, on average, a higher frequency within *Corynebacterium* phages (38%, 8/21) compared to other identified host genera (Fig. 3A). While most of the phage genomes carry only one copy of the *whiB* gene, 182 phages that infect members of the genera *Mycobacterium*, *Propionibacterium*, and *Gordonia* encode two homologs (Fig. S3). Moreover, the distribution of WhiB-encoding phages showed multiple clusters with 100% presence of representative members according to assigned clusters. This observation indicated that closely related phages tend to have a higher chance of harboring such genes (Fig. S4). Calculation of protein pairwise identities revealed a rather low sequence identity of 25 to 48% (Fig. S5), suggesting different functions in the phage circuitry. Moreover, a comparison of the global pairwise identity of WhiB proteins from phages, bacteria, and plasmids

displayed a high variability of phage-encoded proteins, suggesting further functional adaptation (Fig. 3B).

Overall, we observed a higher abundance of *whiB* genes in temperate phages compared to virulent phages (Fig. 3C). Phage genomes encoding this gene but without lifestyle information were not included in this analysis. The comparative analysis with respect to GC content and genome size showed, on average, a lower GC content but an increased overall genome size for *whiB*-encoding temperate and virulent phages (Fig. 3D and E).

**Lsr2-like proteins are enriched in genomes of temperate phages featuring a lower GC content.** Lsr2-like genes were detected in 11% (354/2951) of the total analyzed phage genomes and are distributed over 430 putative loci. Phages encoding Lsr2-like proteins belong to five different host genera, including *Mycobacterium*, *Gordonia*, *Streptomyces*, *Arthrobacter*, and *Microbacterium* (Fig. 4A). Overall, *lsr2* genes were found to be the most abundant in the genomes of phages that infect *Streptomyces*, with approximately 35% of *Streptomyces* phages encoding Lsr2 homologs (20). Phages encoding more than one copy of a *lsr2* gene were found in all host genera except *Microbacterium* (76/354 phage genomes; Fig. S6). In comparison to WhiB, Lsr2-encoding phages also showed a similar distribution pattern according to the known clusters. More than 50% of these clusters showed a 100% presence of phage representative members, suggesting high relatedness between the genomes (Fig. S7). The determination of protein pairwise identities between copies encoded in the same genome revealed significant variation. Interestingly, *Arthrobacter* and *Streptomyces* phage Lsr2 protein copies shared 100% of sequence identity, which is because the corresponding genes are located within the direct terminal repeats (DTR) of these phage genomes (29) (Fig. S8). In contrast, *Mycobacterium* and *Gordonia* sequences share less than 40% of identity between protein copies, suggesting independent gene transfer events and presumably expression of proteins with different functions. Similar to WhiB, phage-encoded Lsr2-like proteins featured the lowest pairwise identity compared to the bacterial- or plasmid-encoded Lsr2 proteins (Fig. 4B).

Analysis of the distribution of Lsr2-encoding phages according to their lifestyle (normalized to group size) showed a three times higher abundancy in temperate phages (Fig. 4C). In agreement with previous studies (20), further analysis revealed a lower GC content on average and a bigger genome size for Lsr2-encoding phages (Fig. 4D and E).

**Neighboring genomic context analyses.** To gain insight into the potential function and origin of phage-encoded WhiB and Lsr2-like proteins, we analyzed the neighboring genomic context in actinobacteriophage genomes, reasoning that gene families enriched in the vicinity of *whiB* and *lsr2* may illuminate their function. For this purpose, we extracted the 5 kb upstream and downstream regions around the *whiB* and *lsr2* genomic loci in actinobacteriophage genomes. These 10 kb regions were centered on either *lsr2* or *whiB* and are referred to as 'extended genomic loci'. All extended genomic loci were then functionally annotated using an automated pipeline (Prokka v. 1.11 [30]) combined with different custom databases (see the Materials and Methods). Overall, this analysis showed a highly syntenic, but cluster-specific, organization of neighboring genes (Fig. 5A to C and Table S3 and S4) (4).

For *whiB* extended genomic loci, domains found in more than 50% of the sequences for each cluster are listed in Table S4. According to our analyses, the most abundant domains are DUF732 (37.9%) and PHA02446 (29.7%) which are associated with uncharacterized or hypothetical proteins as well as HTH_7 (26.3%), SSB (25.2%), and Phage_integrase (19.8%) domains.

In the following, we focused on representative members of the clusters EK, K, and T for visualization of genomic synteny at the *whiB* extended genomic loci (Fig. 5A to C). Phages in the cluster EK infect *Microbacterium foliorum* and share domains associated with DNA replication in the 10 kb region surrounding *whiB* (Fig. 5A), including genes encoding a primase/polymerase (64.3% of the sequences in this cluster), DNA primases (85.7%), DNA helicases (50%), and DNA polymerase I (100%) as well as a Cas4 family exonuclease (100%) and NrdH-like glutaredoxin (37.5%).

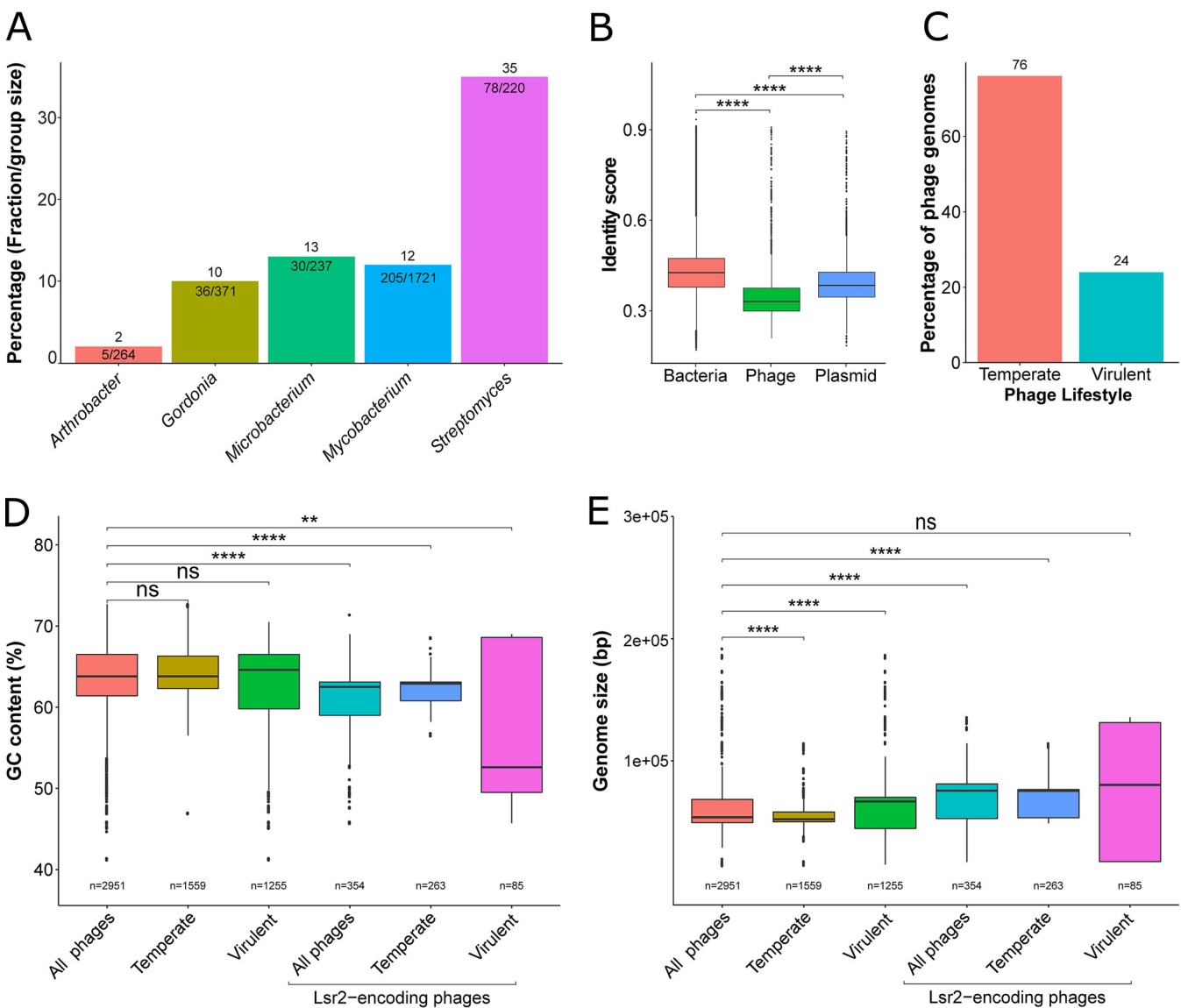

**FIG 4** Lsr2-like proteins are enriched in genomes of temperate phages featuring a lower GC content. (A) Bar plot represents the actual count and proportion of Lsr2-encoding phages according to host genus. (B) Global pairwise identity between Lsr2 amino acid sequences encoded by phage, bacterial, and plasmid phage genomes. (C) Distribution proportion of Lsr2-encoding phages is displayed according to phage lifestyle (temperate and virulent actinophages). (D) Box plot presenting the GC content of actinobacteriophages depending on their lifestyle and the presence of a *lsr2* locus. (E) Box plot presenting the genome size of actinobacteriophages depending on their lifestyle and the presence of a *lsr2* locus. Level of significance is indicated based on the *P value* (ns, not significant; **, $P < 0.01$; ****, $P < 0.0001$).

Based on the *whiB* extended genomic loci context found in the cluster T phages, we hypothesized a role of WhiB in lysis-lysogeny decision making (Fig. 5C). Domains found in the ~10 kb region surrounding *whiB* in those sequences were Phage_ integrase (100% of the sequences), HTH_17 representing excisionases (100%), and DUF3799, which is associated with exonucleases (100%), as well as the COG1476 (100%) and HTH_3 (85.7%) domains associated with Cro/CI family transcriptional regulators. However, many genes in the direct vicinity of *whiB* code for hypothetical proteins (Fig. 5A and C). Therefore, deducing the function based on this analysis is difficult and requires further experiments to elucidate the role of phage-encoded WhiB proteins in the phage life cycle.

Phages in cluster K (96 sequences encoding *whiB*) infect *Mycobacterium smegmatis* (Fig. 5B). Here, an RNase_T domain associated with exonucleases, such as the $\varepsilon$ subunit of DNA polymerase III, were present in 100% of the sequences in this cluster. Other

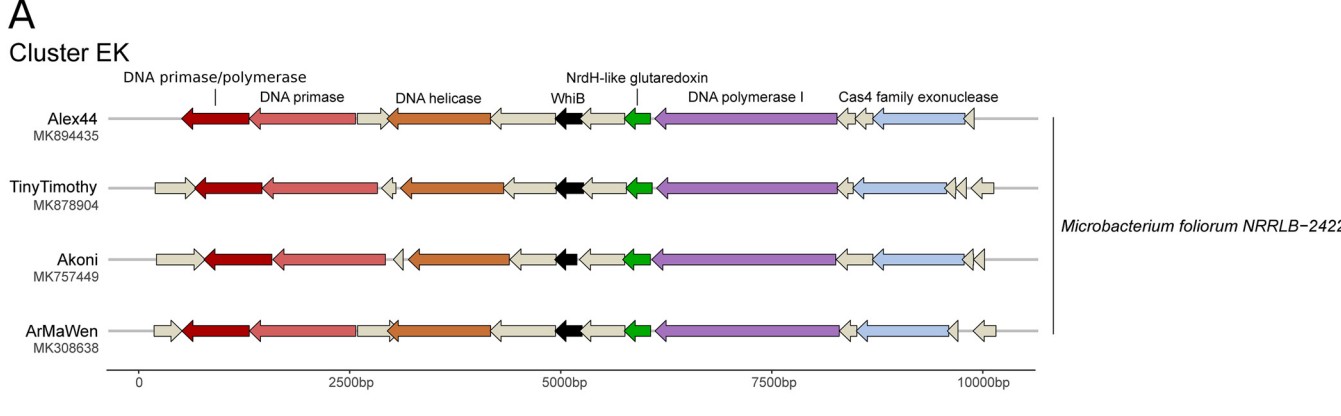

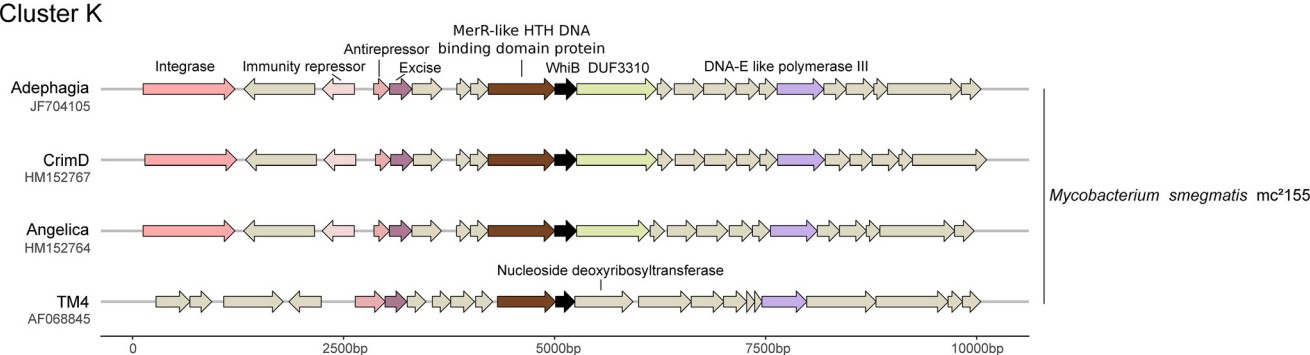

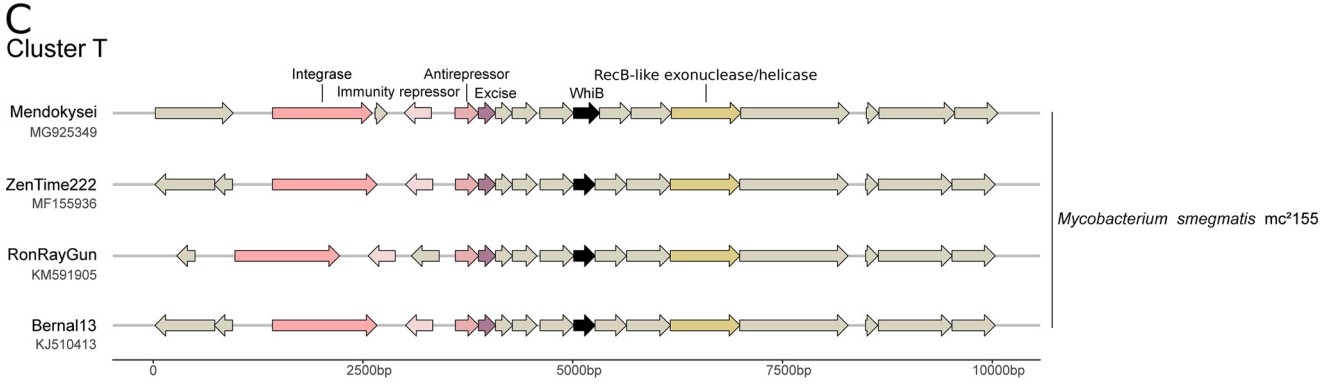

**FIG 5** Syntenic genomic organization of the *whiB* loci in clusters EK, K, and T phages. (A to C) Gene synteny plot showing the 5 kb regions flanking *whiB* across phage genomes of the clusters EK (A), K (B), and T (C) following the cluster assignment determined by phagesDB (4).

abundant domains include Phage_integrase (95.8%) and HTH_17 (84.4%), which is associated with DNA-binding of excisionases. One member of this cluster, TM4 (AF068845) is the only WhiB-encoding bacteriophage investigated for the biological function of WhiB during infection. Rybniker et al. (23) reported WhiB$_{TM4}$ as a dominant, negative regulator that inhibited the gene expression of the host *whiB2* gene. Furthermore, a TM4-resistant phenotype that inhibited secondary infections was observed, pointing toward a function of WhiB in a potential superinfection exclusion mechanism (23). However, synteny of the neighboring genes in the TM4 phage was slightly different from most of the other cluster K sequences because it, for example, lacks the Phage_integrase domain in this region. A recent study focusing on pro-phage-mediated viral defense suggested that the WhiB of temperate phage Tweety (cluster F) was targeted by the heterotypic prophage-mediated defense. Phages lacking the intact protein (by single base substitutions resulting in stop codons) were able to escape the defense mechanism and successfully lyse *M. smegmatis* lysogens (24).

Next, we investigated the gene synteny and GC content within the *lsr2* phage extended genomic loci (Fig. 6 and Table S5 and S6). Similar to *whiB*, we counted all domains detected in the extended genomic loci of *lsr2* and indicated their relative abundance in the different clusters (Table S5 and S6). When we looked for shared domains across different phage clusters, it seemed that the proteins involved in lysogeny (integrase domains: Phage_int_SAM_4, resolvase, and Phage_integrase; immunity repressor: HTH_3; antirepressor domain: HTH_17) and nucleotide metabolism and modification (glutaredoxin domains: Glutaredoxin and Glutaredoxin-like_protein_NrdH; DNA-sulfur modification-associated protein DndB domain) were enriched in the areas surrounding Lsr2. However, it is worth noting that there are marked differences between clusters, and no domain was found in all clusters.

Further, we looked in more detail at the genomic organization of *lsr2* extended genomic loci in members of selected clusters: clusters A, BD, and J (Fig. 6). Interestingly, in cluster A, Lsr2 was found between a module of structural proteins (minor tail proteins) and an operon encoding homologs of the ParA and ParB proteins (Fig. 6A). ParA and ParB are proteins involved in the partitioning of plasmids and bacterial chromosomes during cell division. Additionally, mycobacteriophages of cluster A that lack an integrase were shown to use the *parABS* system to promote maintenance of their DNA as extrachromosally replicating entities (31, 32).

In cluster BD, Lsr2 is found upstream of the integrase and is close to a ribonucleotide reductase and a thymidylate synthase, suggesting a role of Lsr2 in lysogeny and/or nucleotide biosynthesis (Fig. 6B).

Members of cluster J have two copies of *lsr2* (Fig. 6C). The first copy is close to the gene encoding a glutaredoxin homolog and a putative pyrophosphorylase, further highlighting the potential connection of Lsr2 with nucleotide biosynthesis. Intriguingly, the second copy of Lsr2 consists of a fusion of the Ku protein with the C-terminal, DNA-binding domain of Lsr2. More precisely, the first 250 amino acids of this fusion protein are highly homologous with the Ku protein of *Mycobacterium smegmatis*, but the last 30 amino acids show relatively high identity with Lsr2, including the 'AT-hook' RGR motif (16). Ku binds to DNA double-strand break ends and is required for the nonhomologous end joining (NHEJ) pathway of DNA repair in all branches of cellular life (33). Interestingly, two of such Ku-Lsr2 fusion proteins encoded in members of the J cluster phages (Omega and Corndog) were shown to enable circularization of the incoming phage genomes (10), but the presence of the Lsr2 domain was not recognized nor was its contribution investigated.

Previous reports emphasized the preferential binding of Lsr2-like proteins to AT-rich genomic regions in bacterial genomes (15–17). More precisely, the preferential binding was to regions that featured a distinct drop in GC profile close to the transcription start site (19). Inspection of GC profiles in phage genomes at the *lsr2* genomic loci revealed a distinct drop in GC content in the *lsr2* upstream region in members of clusters A, BD, and J, suggesting autoregulation of Lsr2 in these phages (Fig. 6 and Fig. S9). In cluster A phages, an additional drop was observed upstream of the *parAB* operon, which probably corresponds to the *ori* region of these phages. As mentioned above, the *parABS* system is used by cluster A temperate phages that lack an integration cassette to promote their maintenance as extrachromosomal prophages (31). The *ori* region upstream of the *parAB* operon was found to be required for prophage stability. The localization of *lsr2* in this genomic context as well as the distinct drop in GC content upstream of *lsr2* and in the *ori* region could hint towards a role for Lsr2-like proteins in prophage maintenance in cluster A phages. Involvement in the maintenance of lysogeny was previously described for the Lsr2-like xenogeneic silencing protein CgpS in *C. glutamicum* (15, 19). Here, interference with CgpS binding resulted in prophage induction and, consequently, to cell death.

Remarkably, in *Streptomyces* phages belonging to the BE cluster, the *lsr2* genes are located within the direct terminal repeat regions. The direct terminal repeats of BE cluster phages (10 to 12 kbp) are the longest found in actinobacteriophages to date

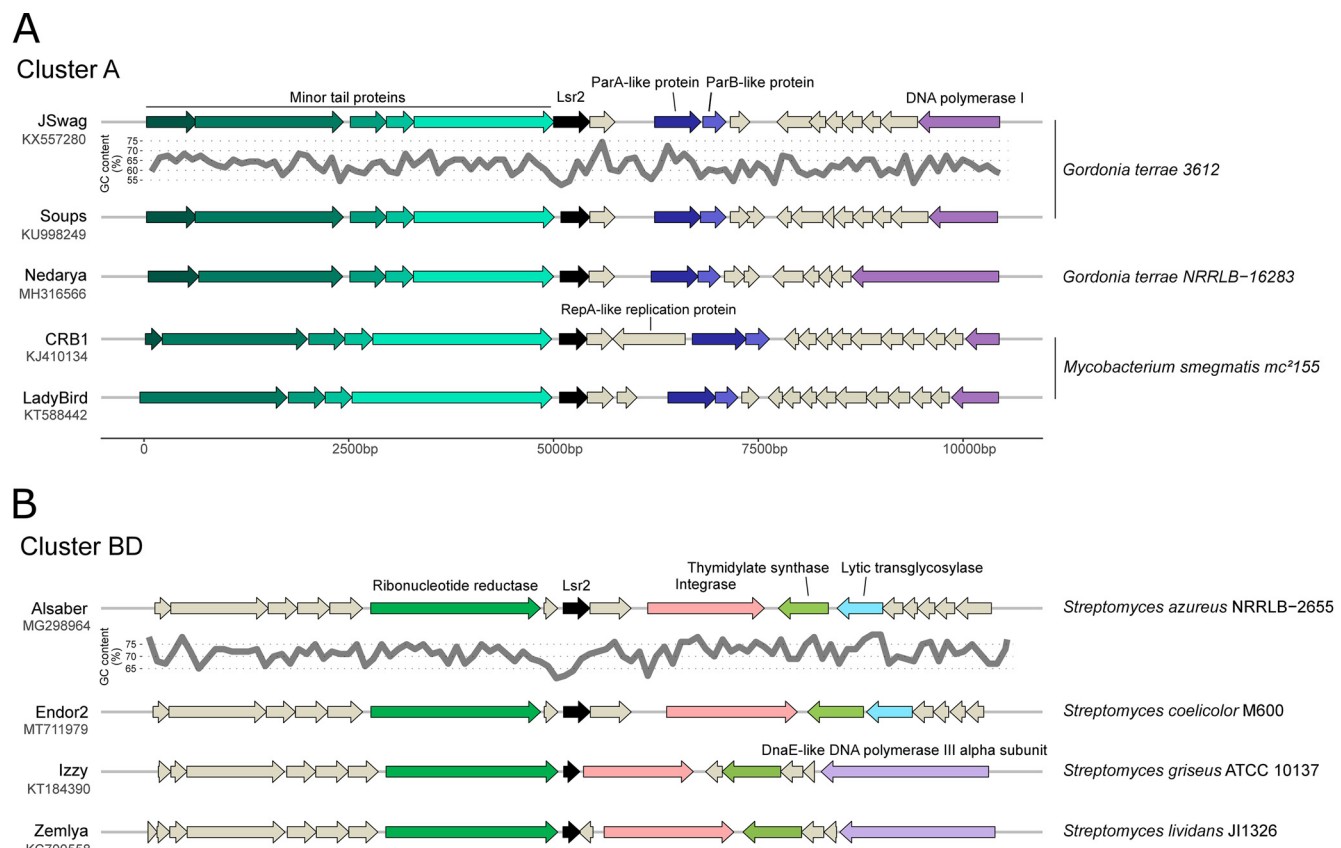

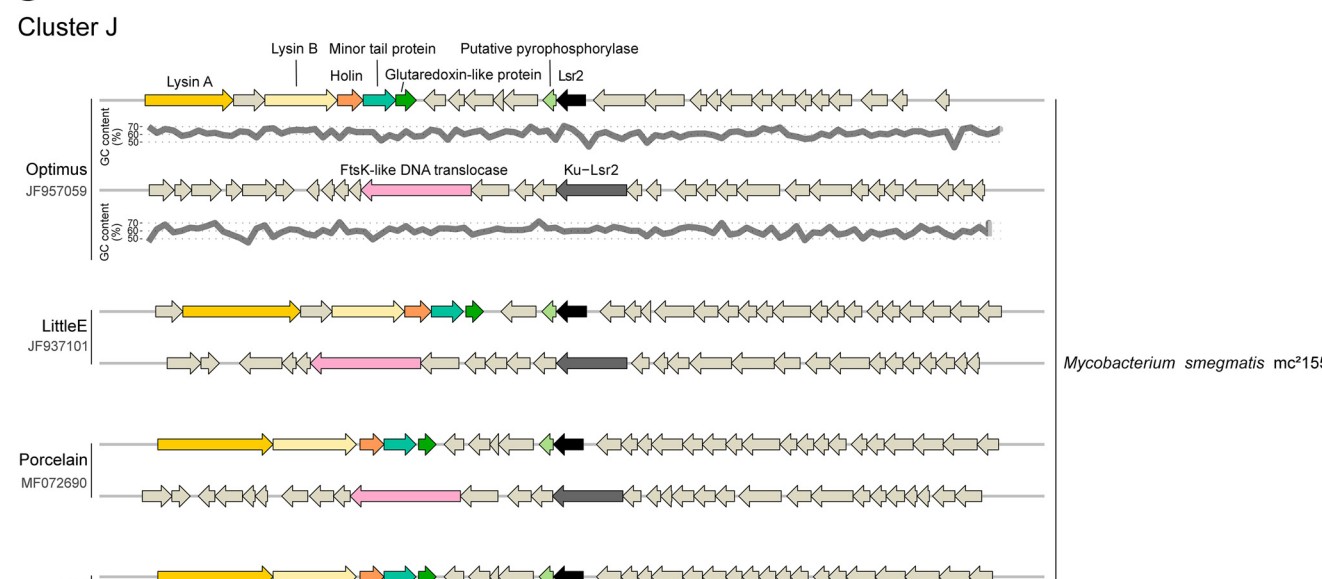

**FIG 6** Syntenic genomic organization of the *lsr2* loci in cluster A, BD, and J phages. (A to C) Gene synteny plot showing the 5 kb regions flanking *lsr2* across randomly selected representative phage genomes of cluster A (A), BD (B), and J (C) following the cluster assignment determined by phagesDB (4). GC content is displayed below the gene synteny plot for the one representative genome per cluster.

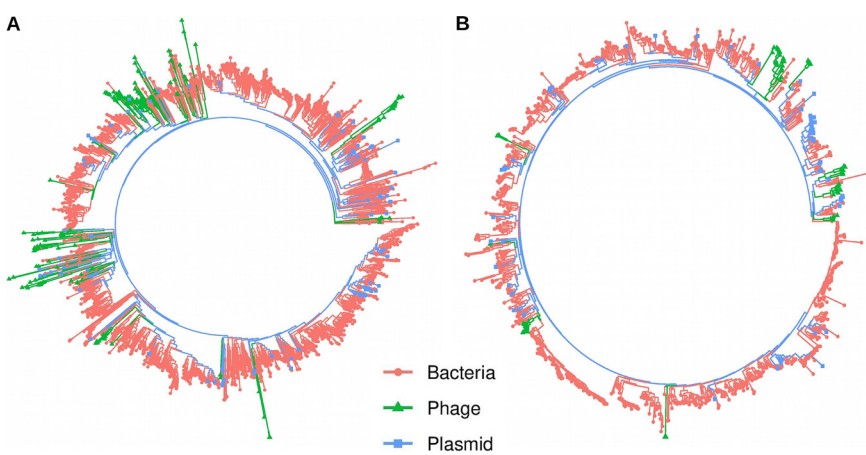

**FIG 7** Phylogenetic analysis revealed complex evolutionary histories of the phage-encoded Lsr2 and WhiB homologs. (A) Unrooted maximum likelihood phylogenetic tree constructed using > 2300 WhiB protein sequences obtained from phage (green), bacteria (red), and plasmid (blue) genomes. Sequences are color-coded based on source groups (bacteria, phage, and plasmid). (B) Unrooted maximum likelihood phylogenetic tree constructed using >950 Lsr2 protein sequences obtained from phage, bacteria, and plasmid genomes.

(29). The genome alignment and GC content comparison of the *lsr2* extended genomic loci from *Streptomyces* phage Coruscant against phage Mildred21 (Fig. S10) showed a significantly higher GC content of the direct repeat region (54.4%) compared to the rest of the genome (47.0%). This finding suggests a block transfer of this ~12 kb region from a phage or bacteria that possessed a distinctively GC-rich genome. However, this large genomic region does not map to any phage or host genome, suggesting acquisition from a bacterial host that has not yet been sequenced.

**Phylogenetic analysis reveals complex evolutionary histories of phage-encoded Lsr2 and WhiB homologs.** The phylogenetic relationship was established using the identified WhiB and Lsr2 protein sequences from actinobacteriophages, plasmids, and bacterial genomes. Instead of using all identified homologs, we separately clustered the protein sequences from bacteria, phage, and plasmid genomes at the 90% amino acid identity threshold using UCLUST (34). The longest sequence was selected as a representative from each resulting cluster and used for further analyses.

The WhiB and Lsr2 maximum likelihood, unrooted phylogenetic tree that was constructed with the high quality protein sequence alignment showed a complex evolutionary history. In the WhiB phylogeny, most phage-encoded sequences form a paraphyletic group and fall within bacterial clades, suggesting viruses acquired genes from their host or that this occurred from independent gene transfers from different sources (Fig. 7A). Zooming into the WhiB subclades showed different evolutionary patterns. One subclade showed gene transfer by *Streptomyces* phages from their host bacteria (Fig. S11). Another subclade displayed gene transfer events within *Mycobacterium* phages (viruses acquired genes from their host; Fig. S12). Here, one bacterial sequence (*NZ_CP034072_1_3673830* to *NZ_CP034072_1_3675228*) deeply clustered within temperate *Mycobacterium* phages. Further inspection revealed that this bacteria-encoded *whiB* gene is located within a prophage region, as determined using the PHASTER online webserver (data not shown) (35). Overall, phage encoded WhiB proteins formed clusters according to the phage lifestyle (Fig. S11 and S12).

Similarly, Lsr2 phylogenetic analysis also showed the paraphyletic grouping of phage sequences within bacterial branches, suggesting acquisition of this gene from their hosts or due to multiple independent acquisition events (Fig. 7B). Zooming in on the phylogenetic groups revealed diverse patterns. One subclade showed gene acquisition in temperate *Streptomyces* and *Gordonia* phages from their host (Fig. S13 and S14). Another subclade showed gene transfer events between unrelated phages (*Mycobacterium* phages to *Gordonia* phages; Fig. S15). However, it is important to

mention that retracing the history of gene transfer events is sometimes impossible, especially between unrelated phages. Moreover, multiple bacterial sequences also form paraphyletic clades for both gene-based phylogenies, reflecting gene exchanges between diverse bacterial species. Overall, *whiB* and *lsr2* gene-based phylogenies revealed complex evolutionary histories. The scattering of the actinobacteriophage sequences on both trees may suggest multiple independent acquisitions of genes by phages from their hosts and related viruses, in accordance with previous reports (20, 36) (Fig. 7A and B).

WhiB- and Lsr2-like proteins are widespread in actinobacteria. Therefore, it is no surprise that phages have taken advantage of their properties. Our analysis revealed that WhiB proteins exist in more than 23% of the sequenced actinobacteriophage genomes, which is similar to the widespread Cro/CI family of phage regulators that have been identified in approximately 20% of the genomes. Previous studies revealed an interplay with the host-encoded WhiB proteins (23), and it is likely that their phage-encoded counterparts may manipulate the network of their host and have an important influence on central processes of their host, such as cellular development, antibiotic production, and/or resistance (21).

Lsr2-like proteins have already been described as xenogeneic silencers that repress the expression of 'foreign', AT-rich DNA elements in host genomes (16, 17, 20, 37). Because foreign DNA in the form of viruses is one of the most primordial threats in the evolution of life, the role of Lsr2-like proteins in the control of the phage gene expression is probably ancient. This hypothesis is also supported by the high variability in amino acid sequence and high conservation of secondary structure of Lsr2 proteins (15). Considering the genomic synteny of *lsr2* in phage genomes as well as the results of previous studies, we postulate a role of phage-encoded Lsr2-like proteins in maintaining the lysogenic state of the prophage. However, expression of a different xenogeneic silencer has also been demonstrated to strongly interfere with the formation of the oligomeric nucleoprotein complex required for silencing and may, therefore, also represent a strategy of virulent phages to defend against silencing upon infection of the host (20). An interesting example is provided by the H-NST protein, a truncated H-NS variant lacking the DNA-binding domain, which was shown to antagonize H-NS in enteropathogenic and uropathogenic *E. coli* (38).

For a comprehensive understanding, experimental approaches are needed to decipher the likely diverse roles of WhiB and Lsr2 proteins in the control of the phage life cycle and their interplay with host regulatory networks. Considering the unique regulatory equipment of actinobacteriophages, future studies will not only deliver fundamental new insights into their regulation but also the minimalist elegance of phage regulatory circuits provides constant inspiration for the fields of synthetic biology and metabolic engineering.

## MATERIALS AND METHODS

**Data collection.** The list of 3396 complete actinobacteriophage available genomes was retrieved on 14 May 2020 from PhagesDB (4). The 2946/3396 complete actinobacteriophage genomes with valid NCBI accession numbers were downloaded with the list information using a NCBI-genome-download python script (https://github.com/kblin/ncbi-genome-download) from the NCBI file transfer protocol (FTP) site. Additionally, five recently sequenced *Streptomyces* phages (accession numbers: MT711976, MT711977, MT711975, MT711978, and MT711979) (5) were included for the current analysis. In total, the 2951 actinobacteriophage genomes that were used in the current study were included in the supplementary information (Table S1). The 2381 complete actinobacterial genomes and plasmid sequences were downloaded from the NCBI FTP site.

**Identification of regulatory gene domains within actinobacteriophage genomes.** Genome-wide distributions of conserved protein domains were determined using the nucleotide sequences of 2951 actinobacteriophages genomes as a query against CDD database using RPS-BLAST version (BLAST 2.2.31+) with an E-value of 0.001 (25). The list of putative transcription factors or regulatory gene information was extracted in tabular format from the three different databases (P2TF [26], Pfam [27], and Phamerator [28]). The regulatory protein domain unique IDs were mapped and filtered from the genome-wide predicted protein domains across the actinobacteriophage genomes using a custom-designed R script. Filtered domains were manually curated for the valid transcription factors based on published literature (Table S2). Lastly, only the domains encoded by more than 10 phage genomes were

included for further analysis to represent the most abundant and widely distributed domains. The density and correlation analysis of identified transcription factors protein domains that were identified inside the actinobacteriophage genomes was displayed using the R packages Superheat (39) and corrplot v. 0.84 (40).

**Whole-genome phylogeny.** We used an alignment-free approach based on a machine learning perspective to estimate the phylogenetic tree with the 2951 actinobacteriophage genomes using an R script (https://bioinformaticshome.com/bioinformatics_tutorials/R/phylogeny_estimation.html). Initially, the script looks for k-mer frequency profiles within the different phage genomes. Pairwise comparisons were then performed to calculate the distance between k-mer pairs using the Jensen-Shannon divergence measure (41). Finally, a phylogenetic tree was obtained after the clustering of the output distance matrix using the neighbor-joining algorithm. The phylogenetic tree visualization and mapping of *whiB* and *lsr2* domains were displayed using the R package ggtree v. 2.0.1 (42).

**WhiB and Lsr2 sequence detection pipeline.** The complete WhiB and Lsr2 sequence homologs were collected independently using a custom design pipeline following a similar approach used in previous studies (43, 44). The TBLASTN searches (45) were conducted using a known set of WhiB and Lsr2 protein sequence libraries as the query against the targeted 2951 actinobacteriophage complete nucleotide sequences with an E-value of $10^{-5}$. The identified overlapping genomic loci hits were merged using SAMtools v. 1.7 (46). To ensure the complete gene sequence length, the *whiB* and *lsr2* genomic loci hits were extended by 500 bp and 5 kbp upstream and downstream, respectively. The extended loci were then translated into proteins using the Emboss translate program. The predicted proteins were compared against the WhiB and Lsr2 sequence libraries using BLASTp (45) with and E-value of $10^{-5}$ to exclude false positives from the analysis. The proteins with the best BLAST hits against the WhiB and Lsr2 proteins were retained and used for further analysis. Besides screening the phage genomes, the same steps of the pipeline were used to identify the WhiB and Lsr2 sequence homologs inside the 2381 complete host actinobacterial genomes and their plasmid sequences.

**Sequence alignment and phylogenetic analysis of WhiB and Lsr2 proteins.** Complete WhiB and Lsr2 protein homologous sequences were collected from phage, bacteria, and plasmid genomes using the custom-designed pipeline. To avoid identical sequences in the analysis, the identified protein sequences for both genes in each source organism (phage, bacteria, and plasmid genomes) were separately clustered at 90% identity using UCLUST (34). The longest sequence was then selected from each resulting cluster as a representative for the sequence alignment. The protein sequences were aligned using the MAFFT v. 7 online server with default parameters (47). The output of multiple sequence alignments was manually curated and checked for the best fit evolutionary model using ProtTest v. 3.4.2 (48). The maximum likelihood-like phylogeny for each gene was constructed separately with the identified optimal model using FastTree v. 2.1.10 (49). The phylogenetic tree was visualized using the R package ggtree v. 2.0.1 (42).

**Neighboring genomic context analysis.** The 5 kb flanking regions extended upstream and downstream around the *whiB* and *lsr2* genomic loci hits were extracted in FASTA format using the previously mentioned homologous detection pipeline. These regions were annotated using the Prokka 1.11 pipeline (30) with the help of multiple databases with different databases (CDD, Conserved Domain Database (30); pVOGs, Prokaryotic Virus Orthologous Groups (50); NCBI viral proteins). The annotated extended regions for both genes were used for gene synteny visualization using the R package Gggenes v. 0.4.0 (https://github.com/wilkox/gggenes). The quantitative gene domain distribution within the annotated extended regions was calculated using a custom R script. The GC content across the *lsr2* extended region was calculated according to a 100 bp window using a Perl script. The GC plot was obtained using ggplot2 v. 3.2.1 in R (51).

**Comparative analyses of WhiB and Lsr2-like proteins.** Comparative analyses the distribution, lifestyle, GC content, and genome size variation of phages containing *whiB* and *lsr2*-like genes were conducted in R using different packages (ggplot2 v. 3.2.1 [51], rstatix v. 0.6.0, gplots v. 3.0.1.2, tidyverse v. 1.3.0, and ggpubr v. 0.4.0). The pairwise identity scores between the WhiB and Lsr2 protein sequences obtained from the bacterial, phage, and plasmid genomes were calculated with MAFFT v. 7.310 (47) using the sequence demarcation tool (SDT) in Python (52). The output of the pairwise identity scores were displayed using the R package ggplot2 v. 3.2.1 (51). Statistically significant differences between the groups were determined based on pairwise Wilcoxon tests using R package rstatix v. 0.6.0.

**Data availability.** The complete sequences of phage genomes used in the current study are available on the PhagesDB website (4). The identified phage encoded WhiB and Lsr2 sequences, phylogenetic trees, and R scripts used for the data analysis are available on a Github repository (https://github.com/sharmavikas3529/Actinophage-regulators-data.git).

## SUPPLEMENTAL MATERIAL

Supplemental material is available online only.
**SUPPLEMENTAL FILE 1**, PDF file, 1.4 MB.
**SUPPLEMENTAL FILE 2**, XLSX file, 0.3 MB.

## ACKNOWLEDGMENTS

We thank the students and faculty members of the SEA-PHAGES program for their contributions to phage discovery and genomics and the SEA-PHAGES program for permission to use the unpublished data compiled at https://phagesdb.org. We especially

thank Graham Hatfull for his expert comments and suggestions. We thank the European Research Council (ERC Starting Grant, grant number 757563) for financial support.

We declare no conflicts of interest.

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
