## [Reviewer comments · Microbiology Spectrum]

Microbiology Spectrum

Phylogenetic distribution of WhiB- and Lsr2-type regulators in actinobacteriophage genomes

Vikas Sharma, Ael Hardy, Tom Luthe, and Julia Frunzke

Corresponding Author(s): Julia Frunzke, Forschungszentrum Jülich GmbH

Review Timeline:

Submission Date:	August 21, 2021
Editorial Decision:	October 11, 2021
Revision Received:	October 25, 2021
Accepted:	October 27, 2021

Editor: Alison Sinclair

Reviewer(s): Disclosure of reviewer identity is with reference to reviewer comments included in decision letter(s). The following individuals involved in review of your submission have agreed to reveal their identity: Simon White (Reviewer #1)

Transaction Report:

DOI: <https://doi.org/10.1128/Spectrum.00727-21>

October 11, 2021

Prof. Julia Frunzke
Forschungszentrum Jülich GmbH
Institute of Bio- and Geosciences, IBG-1: Biotechnology
Leo-Brandt-Str.
Jülich 52425
Germany

Re: Spectrum00727-21 (Phylogenetic distribution of WhiB- and Lsr2-type regulators in actinobacteriophage genomes)

Dear Prof. Julia Frunzke:

Thank you for addressing the concerns of the reviewers for the previous submission. I requested one review for Microbiology Spectrum and this is positive, with a few minor suggestions. Please could you consider these in your final revision.

Thank you for submitting your manuscript to Microbiology Spectrum. When submitting the revised version of your paper, please provide (1) point-by-point responses to the issues raised by the reviewers as file type "Response to Reviewers," not in your cover letter, and (2) a PDF file that indicates the changes from the original submission (by highlighting or underlining the changes) as file type "Marked Up Manuscript - For Review Only". Please use this link to submit your revised manuscript - we strongly recommend that you submit your paper within the next 60 days or reach out to me. Detailed information on submitting your revised paper are below.

Link Not Available

Sincerely,

Alison Sinclair

Journals Department
Reviewer comments:

Reviewer #1 (Comments for the Author):

Sharma et al have written an excellent paper on the Lsr2-type and WhiB-type regulators in the actinobacteriophages. Really interesting paper. Minor comments.

Minor comments:

Line 60. There are more hosts used in the SEA-PHAGES program than the ones listed, for example Microbacterium accounts for over 400 sequenced phages. In table S1 it is clear you have used all of the SEA-PHAGES hosts so you need to increase your list on this line.

Line 64. Not sure "Remarkably" is appropriate here. The method used to isolate the genome in the SEA-PHAGES program would only purify dsDNA so it isn't surprising that all the actinobacteriophages isolated from the program are dsDNA.

Line 76. There are far more than 24 distinct clusters. Including sub clusters that number goes well over 100.

Line 77. Are they "Highly divergent phages"? Not sure what is meant by this line. I've always assumed it is a sampling issue and that singletons are just a future cluster. The way it is written suggests that they are somehow different than other phages. I'd suggest that the line be "Phages without close relatives are assigned as singletons".

Line 86. "make a decision" is inappropriate language for viruses. We all do this when talking about viruses (including myself). Viruses aren't capable of deciding anything. They also don't own the host cell ("infection of their host cell"). Maybe change it to "temperate phages follow one of two lifecycle paths upon infection of the host cell"?

Line 95. Perhaps making it clearer that most tailed bacteriophages studied to date infect the Proteobacteria? I think the work in this paper is made more interesting because it is looking at the phages of a completely different bacterial phylum.

How is Table S1 organized? Looks almost random to me. Is almost by accession number but isn't. I'd suggest ordering them based on cluster name.

I'm not going to ask you do it for this paper but in the future (if you do similar analyses) could you use more than just accession number in table S2. It would have been useful to include phage name and cluster. Would make possible cluster patterns in table S2 more obvious to a reader.

Line 138. "10% of the analyzed phage genomes". I couldn't see anything about cluster bias. A and F clusters are huge and could well account for 10% of phage genomes.

Figure 2. Is there a key for the color coding of the clusters? I think this needs to be included.

Line 468. Did you mean secondary structure? Just the helicies and beta sheets? Or did you mean the overall fold of the proteins, the tertiary structure?

Line 489 That link does not work for me.

Line 569. That link does not work for me.

References need double checking. Page numbers often missing. I've put some that I found below but double check all. Some have doi and some don't.

582. Has no journal, page number, etc.

596. Why does it have doi?

626: No volume or page number

668. No volume or page number

670 No volume or page number

673 No volume or page number

Staff Comments:

Preparing Revision Guidelines

- Point-by-point responses to the issues raised by the reviewers in a file named "Response to Reviewers," NOT IN YOUR COVER LETTER.
- Upload a compare copy of the manuscript (without figures) as a "Marked-Up Manuscript" file.
- Each figure must be uploaded as a separate file, and any multipanel figures must be assembled into one file.

- Manuscript: A .DOC version of the revised manuscript
- Figures: Editable, high-resolution, individual figure files are required at revision, TIFF or EPS files are preferred

Please return the manuscript within 60 days; if you cannot complete the modification within this time period, please contact me. If you do not wish to modify the manuscript and prefer to submit it to another journal, please notify me of your decision immediately so that the manuscript may be formally withdrawn from consideration by Microbiology Spectrum.

Reviewer #1 (Comments for the Author):

Sharma et al have written an excellent paper on the Lsr2-type and WhiB-type regulators in the actinobacteriophages. Really interesting paper. Minor comments.

Answer: We are thankful for your comments and suggestions. We have modified the manuscript according to these comments and suggestions.

Line 60. There are more hosts used in the SEA-PHAGES program than the ones listed, for example *Microbacterium* accounts for over 400 sequenced phages. In table S1 it is clear you have used all of the SEA-PHAGES hosts so you need to increase your list on this line.

Answer: As per your suggestion, all SEA-PHAGES hosts mentioned in table S1 are now included in the text.

Line 64. Not sure "Remarkably" is appropriate here. The method used to isolate the genome in the SEA-PHAGES program would only purify dsDNA so it isn't surprising that all the actinobacteriophages isolated from the program are dsDNA.

Answer: We replaced "Remarkably" with "Moreover".

Line 76. There are far more than 24 distinct clusters. Including sub clusters that number goes well over 100.

Answer: We do agree with your comment. Additional information about the subclusters is now included in the text.

Line 77. Are they "Highly divergent phages"? Not sure what is meant by this line. I've always assumed it is a sampling issue and that singletons are just a future cluster. The way it is written suggests that they are somehow different than other phages. I'd suggest that the line be "Phages without close relatives are assigned as singletons".

Answer: As per your suggestion, the text is modified in the manuscript.

Line86. "make a decision" is inappropriate language for viruses. We all do this when talking about viruses (including myself). Viruses aren't capable of deciding anything. They also don't own the host cell ("infection of their host cell"). Maybe change it to "temperate phages follow one of two lifecycle paths upon infection of the host cell"?

Answer: As advised, this sentence is modified in the manuscript.

Line 95. Perhaps making it clearer that most tailed bacteriophages studied to date infect the Proteobacteria? I think the work in this paper is made more interesting because it is looking at the phages of a completely different bacterial phylum.

Answer: Thank you for this meaningful remark. We have shortly highlighted the more comprehensive work done with Proteobacteria phages in comparison to actinophages in the text.

How is Table S1 organized? Looks almost random to me. Is almost by accession number but isn't. I'd suggest ordering them based on cluster name.

Answer: Yes, the Table S1 organization is random. However, as per your suggestion, it is now organized according to the cluster names.

I'm not going to ask you do it for this paper but in the future (if you do similar analyses) could you use more than just accession number in table S2. It would have been useful to include phage name and cluster. Would make possible cluster patterns in table S2 more obvious to a reader.

Answer: Thank you very much for your suggestion. In future work, we will include such information in the analysis.

Line 138. "10% of the analyzed phage genomes". I couldn't see anything about cluster bias. A and F clusters are huge and could well account for 10% of phage genomes.

Answer: We do agree with your comment. Therefore, an additional sentence has been amended in the manuscript.

Figure 2. Is there a key for the color coding of the clusters? I think this needs to be included.

Answer: We agree with this comment, but due to more than 100 clusters, it is hard to represent the color-coding keys.

Line 468. Did you mean secondary structure? Just the helicies and beta sheets? Or did you mean the overall fold of the proteins, the tertiary structure?

Answer: Yes, here we mentioned about the overall secondary structural similarity between the CgpS and Lsr2 proteins.

Line 489 That link does not work for me.

Answer: The link was cross-checked and found to be working fine. Please ensure that site is not blocked or try it with another machine.

Line 569. That link does not work for me.

Answer: The link was cross-checked and found to be working fine. Please ensure that site is not blocked or try it with another machine.

References need double checking. Page numbers often missing. I've put some that I found below but double check all. Some have doi and some don't.

Answer: Thank you for noticing these unfortunate mistakes. All references have been cross-checked and corrected with the updated information in the manuscript.

582. Has no journal, page number, etc.

Answer: Journal and page number are now included in the reference.

596. Why does it have doi?

Answer: Thank you for spotting this mistake. The corrected information is now updated in the reference.

626: No volume or page number

Answer: Volume and page number are included in the reference.

668. No volume or page number

Answer: Volume and page number are included in the reference.

670 No volume or page number

Answer: Volume and page number are included in the reference.

673 No volume or page number

Answer: Volume and page number are included in the reference.